# *Staphylococcus aureus* α-Toxin Effect on *Acinetobacter baumannii* Behavior

**DOI:** 10.3390/biology11040570

**Published:** 2022-04-09

**Authors:** Jennifer S. Fernandez, Marisel R. Tuttobene, Sabrina Montaña, Tomás Subils, Virginia Cantera, Andrés Iriarte, Lorena Tuchscherr, Maria Soledad Ramirez

**Affiliations:** 1Center for Applied Biotechnology Studies, Department of Biological Science, California State University Fullerton, Fullerton, CA 92831, USA; jennifer.steph.fernandez@gmail.com (J.S.F.); tuttobene@ibr-conicet.gov.ar (M.R.T.); 2Instituto de Biología Molecular y Celular de Rosario (IBR-CONICET), Rosario S2000, Argentina; 3Laboratorio de Bacteriología Clínica, Departamento de Bioquímica Clínica, Hospital de Clínicas José de San Martín, Facultad de Farmacia y Bioquímica, Buenos Aires C1113, Argentina; sabri.mon@hotmail.com; 4Instituto de Procesos Biotecnológicos y Químicos de Rosario (IPROBYQ, CONICET-UNR), Rosario S2000, Argentina; subils@iprobyq-conicet.gob.ar; 5Laboratorio de Biología Computacional, Departamento de Desarrollo Biotecnológico, Facultad de Medicina, Universidad de la República, Montevideo 11200, Uruguay; vircantera@gmail.com (V.C.); airiarteo@gmail.com (A.I.); 6Institute of Medical Microbiology, Jena University Hospital, 07747 Jena, Germany; lorena.tuchscherrdehauschopp@med.uni-jena.de

**Keywords:** *Acinetobacter baumannii*, *Staphylococcus aureus*, polymicrobial infections, commensals

## Abstract

**Simple Summary:**

Polymicrobial infections, infections that are caused by more than one pathogen, are known to be responsible associated with high mortality rates. The bacterial organisms involved in these infections favor each other, resulting in pathogens’ success. Our work examined the behavioral response of *Acinetobacter baumannii* after exposure to cell-free conditioned media of *Staphylococcus aureus*, two pathogens responsible for severe infections. We focus on the effect of α-toxin, the primary cytotoxic agent released by *S. aureus*, on *A. baumannii’s* behavior. Results indicated that α-toxin contributes to the proliferation and survival of *A. baumannii*. One or more soluble molecules secreted by *S. aureus* can be sensed by *A. baumannii* and trigger diverse responses to adapt to environmental changes. The coexistence between bacteria can result in modifications in their general biology.

**Abstract:**

Polymicrobial infections are more challenging to treat and are recognized as responsible for significant morbidity and mortality. It has been demonstrated that multiple Gram-negative organisms take advantage of the effects of *Staphylococcus aureus* α-toxin on mucosal host defense, resulting in proliferation and dissemination of the co-infecting pathogens. Through phenotypic approaches, we observed a decrease in the motility of *A. baumannii* A118 after exposure to cell-free conditioned media (CFCM) of *S. aureus* strains, USA300 and LS1. However, the motility of *A. baumannii* A118 was increased after exposure to the CFCM of *S. aureus* strains USA300 Δ*hla* and *S. aureus* LSI Δ*agrA*. Hemolytic activity was seen in A118, in the presence of CFCM of *S. aureus* LS1. Further, *A. baumannii* A118 showed an increase in biofilm formation and antibiotic resistance to tetracycline, in the presence of CFCM of *S. aureus* USA300. Transcriptomic analysis of *A. baumannii* A118, with the addition of CFCM from *S. aureus* USA300, was carried out to study *A. baumannii* response to *S. aureus’* released molecules. The RNA-seq data analysis showed a total of 463 differentially expressed genes, associated with a wide variety of functions, such as biofilm formation, virulence, and antibiotic susceptibility, among others. The present results showed that *A. baumannii* can sense and respond to molecules secreted by *S. aureus*. These findings demonstrate that *A. baumannii* may perceive and respond to changes in its environment; specifically, when in the presence of CFCM from *S. aureus*.

## 1. Introduction

Gene expression, affecting virulence traits and metabolic responses, can occur when two or more different species are present in the same environment [1]. During polymicrobial infection, the interaction between the organisms present at the site of infection might significantly contribute to the spread, cause a synergistic interaction, affect antibiotic resistance, and delay the outcome of infections [2,3,4]. This can, in part, explain why polymicrobial infections are more challenging to treat and are also recognized as responsible for significant morbidity and mortality [5]. Co-infection between *Staphylococcus aureus* and *Pseudomonas aeruginosa*, mainly in a cystic fibrosis respiratory infection, has been widely reported [6]. However, co-infections between *S. aureus* and other Gram-negative bacilli have been scarcely reported [7,8]. In our previous work, where we studied the two clinical strains of *S. aureus* and *A. baumannii* recovered from the same site of infection, we observed that there was no significant decrease in the growth of either strain, which indicated that both strains can co-exist at the same site of infection [9]. It has also been reported that during infection, *S. aureus* produces virulence factors, such as cytotoxins, altering the host’s immune response and increasing its survival. Among them, alpha (α)-toxin is the major cytotoxic agent utilized by *S. aureus* [10]. α-toxin is encoded by the gene *hla*, under the regulation of the *agr*, *sarA* and *sae* genes [11]. It has also been shown that *S. aureus* α-toxin potentiates Gram-negative bacterial proliferation, systemic spread, and lethality by preventing acidification of bacteria-containing macrophage phagosomes, thereby reducing the effective killing of both *S. aureus* and Gram-negative bacteria [12]. Considering this evidence, we aimed to explore the effect of this α-toxin on *A. baumannii*’s behavior.

## 2. Materials and Methods

### 2.1. Bacterial Strains and Growth Conditions

*A. baumannii* strains A118 (highly susceptible strain), A42 (intermediately susceptible strain) and AB5075 (strong resistance and highly virulent) were used in the present study [13,14,15,16,17,18,19]. For the generation of the cell-free conditioned media (CFCM) and the zone of clearance (ZOC) assays, the following *S. aureus* wild-type strains and derivate mutants were used: LS1 (MSSA), USA300 (MRSA), LS1 Δ*agrA*, USA300 Δ*agrA*, USA300 Δ*hla* and USA300 Δ*hla*, harboring plasmid pAH5 expressing a wild-type copy of *hla* (USA300Δ*hla* comp) [20,21,22,23].

### 2.2. Cell-Free Conditioned Media (CFCM)

Bacterial strains were grown in several different media (LB, TSB and BHI) for 48 h at 37 °C under shaking conditions (200 rpm). Cultures were then centrifuged, the supernatant was collected and then filtered (0.22 μm). Then, the resulting cell-free conditioned media (CFCM) were stored at −80 °C, as previously described [24].

### 2.3. Motility Assays

Motility agar plates were prepared as previously described [25]. Briefly, A118, A42, and AB5075 cells were cultured in LB broth in the absence or presence of CFCM (50%) and incubated with agitation for 18 h at 37 °C. Then, 4 μL of the overnight culture was pipetted onto the center of a motility agar plate and incubated at 37 °C for 24 h, and the diameter of growth was measured and recorded. Experiments were performed in triplicate.

### 2.4. Hemolytic and Fibrinolytic Activity Assays

*A. baumannii* cells were cultured in LB, TSB and BHI broth with or without 50% CFCM and incubated with agitation for 18 h at 37 °C. Then, for fibrinolytic activity assays, 5 μL of culture was spot seeded on heated plasma agar plates (HPA) [26] and on Tryptic soy agar (TSA) plates supplemented with 5% sheep’s blood (blood agar plates; Hardy Diagnostics, Santa Maria, CA, USA) for hemolytic activity assays [27]. HPA and blood agar plates were incubated for up to 72 h at 37 °C and observed for the presence of characteristic halos. Experiments were performed in triplicate.

### 2.5. Biofilm Formation

*A. baumannii* cells were cultured in LB and TSB broth with or without 50% CFCM and incubated with agitation for 18 h at 37 °C. Quantification of biofilm production in polystyrene wells was carried out using a protocol from previously described method [25]. Briefly, overnight cultures were centrifuged at 5000 rpm at 4 °C for 5 and cell pellets were washed twice with 1X PBS and then re-suspended in 1X PBS. Following this, the optical density at 600 nm (OD600) of each culture was adjusted to 0.9–1.1, vortexed, and diluted 1:100 in LB or TSB broth with or without 50% CFCM before being plated in technical triplicate in a 96-well polystyrene microtiter plate and being incubated at 37 °C for 24 h without agitation. The following day, the OD600 (ODG) was measured using a microplate reader (SpectraMax M3 microplate/ cuvette reader with SoftMax Pro v6 software) to determine the total biomass. Wells were emptied with a vacuum pipette, washed three times with 1X phosphate-buffered saline (PBS), and stained with 1% crystal violet (CV) for 15 m. Excess CV was removed by washing three more times with 1X PBS and the biofilm associated with the CV was solubilized in ethanol acetate (80:20) for 30 m. The OD580 (ODB) was measured using a microplate reader and the results were reported as the ratio of biofilm to total biomass (ODB/ODG). Experiments were performed in triplicate.

### 2.6. Susceptibility Assays

Antibiotic susceptibility assays were performed following the procedures recommended by the CLSI [28], with slight modifications as described by Ramirez et al. [29]. Briefly, Mueller–Hinton agar plates were inoculated with 100 μL of culture of each tested condition (LB or LB with 50% CFCM) after OD adjustment. Then, antimicrobial commercial minimum inhibitory test strips were placed on the plates and the plates were incubated at 37 °C for 18 h. The assays were performed in triplicate.

### 2.7. Zone of Clearance (ZOC) Assays

To generate seeded plates, *A. baumannii* cells were resuspended into 0.2 mL 0.9% NaCl. Then, 15 mL of BHI agar was inoculated with 8 μL of cell suspension and poured into a plate. The plates were then cooled for 30 min and then 25 μL of *S. aureus* strain cell suspensions was spotted onto the seeded plates and allowed to dry for 40 min. The plates were incubated at 28 °C and were examined every 24 h for 120 h. Every 24 h interval, images were recorded and the zones of clearance were measured. The assays were performed in triplicate.

### 2.8. RNA Extraction, Sequencing and Analysis

Overnight cultures of A118 were diluted 1:10 in either fresh LB broth or in 100% CFCM_USA300_ and incubated with agitation for 7 h at 37 °C. RNA was immediately extracted using the TRI REAGENT^®^ Kit (Molecular Research Center, Inc., Cincinnati, OH, USA). Quantification of RNA was performed using a DeNovix DS-11+ Spectrophotometer and qualification was assessed on a 1.2% agarose gel via gel electrophoresis. DNase treatment was performed following the manufacturer’s instructions (Thermo Fisher Scientific, Waltham, MA, USA) and results were quantified as previously described [14]. To confirm sample were free of DNA contamination, PCR amplification of the 16S rDNA gene was performed. Extractions were performed in triplicate with controls performed in parallel. RNA sequencing was outsourced to Otogenetics (Otogenetics Corporation, Atlanta, GA, USA) where they performed ribosomal RNA-depletion using the Ribo-Zero kit (Illumina, San Diego, CA, USA). Construction of the cDNA library was performed with the TruSeq Stranded Total RNA Library Prep kit (Illumina) from three independent replicates per sample. Data collected generated an average of 19.5 million paired-end reads per sample with an average of 19.8 and 18.8 total reads for LB (5 replicates) and CFCM_USA300_ (3 replicates), respectively. Next, 3′-end adapter contaminant trimming was conducted using scythe (available online: github.com/vsbuffalo/scythe (accessed on 18 February 2022). Then, reads were trimmed based on quality using sickle software (available online: github.com/najoshi/sickle (accessed on 18 February 2022) [30] and a phred score of 35 was used as threshold. Reads were mapped against the annotated draft genome of A118 with the function “align” from the R subread package [31]. Mapped reads (averaged 90%) were counted based on the available annotation data using the function “feature Counts”, also from the Rsubread package [31]. Finally, differential expression analysis was carried out with the DESeq 2 package [32]. Genes with an FDR-adjusted *p*-value of <0.05 were considered statistically significantly differentially expressed and subject to further analysis by Artemis Version 16.0.0 (Sanger, Hinxton, UK), Blast2Go Version 4.1.5 (Biobam, Valencia, Spain), NCBI BLAST (National Center for Biotechnology Information, Bethesda, MD, USA), and the Kyoto Encyclopedia of Genes and Genomes (KEGG) (Kyoto Encyclopedia of Genes and Genomes, Kyoto, Japan). RNA-seq data generated as a result of this work has been deposited in SRA with the accession PRJNA791320.

### 2.9. Statistical Analysis

Statistical analysis *t* test or ANOVA followed by Tukey’s multiple comparison test (as appropriate) was performed using GraphPad Prism (GraphPad software, San Diego, CA, USA), and a *p*-value < 0.05 was considered significant.

## 3. Results

### 3.1. Effect on Motility and Biofilm Formation

To evaluate the effect of *S. aureus*-released molecules on *A. baumannii*’s behavior, different phenotypic assays were performed. We began with conducting motility experiments, where it was observed that the addition of CFCM_USA300_, CFCM_USA300 Δ*hla*_ and CFCM_USA300 Δ*hla* comp_, obtained from LB, TSB and BHI, caused a marked decrease in A118 motility, with respect to the control without CFCM (Figure 1A). Moreover, the diameter of motility of A118 also decreased when exposed to CFCM_LS1 Δ*agrA*_ (BHI) (Figure 1A). The *agr* operon is known to directly play a role in the virulence of *S. aureus* [33]. Figure 1A, furthermore, shows that the A42 strain exhibited a decrease in the diameter of motility with the addition of CFCM_USA300_ and CFCM_USA300 Δ*hla* comp_, whereas an increase was observed with the addition of CFCM_USA300 Δ*hla*_ obtained from LB. In contrast, a slight decrease was observed with the addition of these three CFCM acquired from TSB and BHI (Figure 1A).

Moreover, *A. baumannii* was exposed to CFCM of the methicillin-sensitive *S. aureus* LS1 strain and its derivate mutant, Δ*agrA*. For the *A. baumannii* A118 strain, a significantly decreased motility diameter was observed in the presence of CFCM_LS1_ and CFCM_LS1 Δ*agrA*_ obtained from the three media analyzed, in comparison to the control condition (Figure 1B). The A42 strain showed a reduction in motility under CFCM_LS1_ and enhanced motility under the CFCM_LS1 Δ*agrA*_ treatment, obtained from LB and TSB (Figure 1B). Finally, changes in motility were not observed for the *A. baumannii* AB5075 strain, supplemented with different analyzed CFCM (Figure 1A,B).

The analysis of the *A. baumannii* genome sequence has revealed the absence of genes required for flagellar biosynthesis, necessary for swarming motility [34,35]. However, *A. baumannii* exhibits flagellum-independent motility, such as twitching motility and surface-associated motility. To analyze the expression of genes related to motility, twenty-six genes were evaluated by RNA-seq; as a result, five genes related to motility were DEG when *A. baumannii* A118 was grown in the presence of CFCM from *S. aureus* USA300 (Figure 1C).

Biofilm formation is found in various environmental niches and commonly comprises two or more bacterial species [36]. In this work, *A. baumannii* biofilm formation assay was performed in the presence of CFCM obtained from LB and TSB. The results for A118 and A42 strains showed that under CFCM_USA300_ and CFCM_USA300_
_Δ*hla*_ treatments, major biofilm formation was observed, while differences were not observed for AB5075 strains under these CFCM. Lastly, a significant decrease in biofilm formation, in the presence of CFCM_USA300_
_Δ*hla* comp_, was obtained for the three *A. baumannii* strains analyzed (Figure 2A).

### 3.2. Effect of S. aureus-Released Molecules on Other Phenotypes

We also studied other important phenotypes that can be affected by the presence of *S. aureus* and the impact it can have on the pathobiology of *A. baumannii*, such as hemolytic activity and antimicrobial resistance.

The *A. baumannii* strains studied in this work are characterized as not hemolytic. However, hemolytic activity was observed when *A. baumannii* cells were cultivated in the presence of *S. aureus*-released molecules contained in CFCM_LS1_, cultured in LB and BHI (Figure 3A). In contrast, CFCM_LS1 Δ*agrA*_ did not induce hemolytic activity in the *A. baumannii* strain. This result could be expected, as the Agr regulator is the major coordinator of the *hlA* expression in *S. aureus* [37].

Since the *A. baumannii* TU04 strain was reported to have fibrinolytic activity [38], we carried out fibrinolytic activity assays [26]. However, fibrinolytic activity was not observed, either for the *A. baumannii* strain utilized in this study grown in LB or in CFCM tested conditions (data not shown).

*S. aureus*’s effect on antibiotic susceptibility phenotypes revealed no change in the MIC for tetracycline, in any of the CFCM tested conditions for A118. In A42, an increase in tetracycline MIC was observed with CFCM_USA300_ (Appendix A). In addition, in AB5075, changes in the MIC for tetracycline were observed under CFCM_USA300_ and CFCM_USA300 Δ*hla*_
_comp_ treatments (Figure 4A and Appendix A) and for imipenem, in the presence of CFCM_USA300 Δ*hla*_ and CFCM_USA300 Δ*hla*_
_comp_ (Appendix A).

We also tested *S. aureus* and its isogenic mutants for bactericidal activity against *A. baumannii*. ZOC assays for A118 showed an increase in the zone of clearance in the presence of USA300_Δ*hla* comp_ and LS1, after two days, while, after four days, an increase was observed for LS1_Δ*agr*_. In addition, bactericidal activity was found in A42, when challenged with LS1 after day one and with USA300_Δ*hla*_ and USA300_Δ*hla* comp_ after day two, while in AB5075, after day one, a zone of clearance was seen when exposed to USA300, USA300 _Δ*hla*_ comp and LS1_Δ*agr*_. After day four, an effect was also seen when USA300_Δ*hla*_ and LS1 were present (Appendix A).

### 3.3. RNA Sequencing Analysis

In total, 463 significantly differentially expressed genes (DEG) were identified (FDR < 0.05). Among them, 244 were down-regulated and 219 were up-regulated when *A. baumannii* A118 was exposed to CFCM_USA300_ (Appendix A).

From the 244 down-regulated DEGs, 192 were analyzed. The remaining 52 down-regulated DEGs produced hypothetical proteins, protein of unknown functions, uncharacterized proteins, and putative protein-coding genes. The 192 repressed DEGs were categorized into different functions; for example, 89 were related to metabolism, 62 were involved in transport of bio-materials and 18 were related to antibiotic resistance (Figure 4B). As shown in Figure 1C, the *pilP*, *pilM*, *pilQ*, *pilB* and *fimV* genes, related with motility, were also down-regulated. This result was consistent with a reduction in motility observed for the A118 strain in the presence of CFCM_USA300_ (Figure 1A). Moreover, other down-regulated coding genes were related to DNA binding and repair, detoxification, oxidative stress response, acetoin metabolism and transformation.

Within the 219 DEGs up-regulated, 14 were hypothetical proteins and were not further discussed. Important biofilm formation-related genes, such as *csuA/B*, *csuC* and *ompA*, were found (Figure 2B). This result was consistent with a major biofilm formation under CFCM_USA300_ treatment, with respect to the control without CFCM (Figure 2). In addition, it was reported that the action of iron-uptake systems contributes to *A. baumannii* virulence and pathobiology [39]. Transcriptomic analysis of *A. baumannii* A118 showed that the expression of 17 genes related with iron acquisition and metabolic functions was significantly up-regulated in the presence of CFCM_USA300_ (Figure 3B). Furthermore, among other positively regulated coding genes, we identified virulence and Phenylacetic acid metabolism-related functional terms.

## 4. Discussion

*A. baumannii* and *S. aureus* are pathogenic bacteria belonging to the ESKAPE group. They generate different types of infections that are usually difficult to treat [40]. Often, these infections are polymicrobial and pathogens, such as *A. baumannii*, *S. aureus*, *Pseudomonas* spp., *Escherichia* spp., *Klebsiella pneumonia*, among others, can coexist [41,42]. Different pathogens may act synergistically or in succession to mediate polymicrobial infections [9].

Epidemiologic studies identified a transition from *S. aureus* to Gram-negative organisms as the primary pathogens in early- to late-onset ventilator-associated pneumonia [43,44], as well as in cystic fibrosis patients [45]. These results indicate that *S. aureus* colonization could stimulate infection by other bacterial species. In addition, Cohen et al. [12] demonstrated that multiple Gram-negative organisms take advantage of the effects of *S. aureus* α-toxin on mucosal host defense, resulting in proliferation and dissemination of the co-infecting Gram-negative pathogens. In that work, utilizing a murine lung infection model, it was demonstrated that α-toxin enhanced the growth and dissemination of *P. aeruginosa*, *K. pneumonia* and *A. baumanni*, by preventing acidification of bacteria containing macrophage phagosomes [12]. Furthermore, *S. aureus* is the most frequently co-isolated pathogen in diabetic foot ulcer, ranging from soft tissue to bone infections [46]. In our previous report, we analyzed a strain of *A. baumannii* and a strain of *S. aureus* that were both recovered from skin and soft tissues of a diabetic patient [9], and showed that these two pathogens can be causative agents of diabetic foot ulcer and can co-exist in the site. In addition, we found that both strains do not have an effect on one another, whether beneficial or detrimental, showing a state of commensalism between the two. This relationship has been previously observed where both strains have been co-cultured together, with no significant decrease in growth in either clinical strain, and without experiencing statistically significant changes in susceptibility [9].

In this work, we demonstrate that one or more soluble molecules secreted by *S. aureus* can be sensed by *A. baumannii* and trigger different pathogenic responses to adapt to environmental changes. Future research should be carried out to further investigate potential molecular responses in *A. baumannii*. We showed that recognized virulence responses, such as motility and biofilm formation, of *A. baumannii* were variable, depending on the CFCM with which LB broth was supplemented. Using mutant strains in α-toxin and in the master regulator Agr, a decrease in motility was observed for the A118 strain, while the opposite effect was observed for the A42 strain, indicating that the molecular mechanisms that regulate motility are strain-dependent. When *A. baumannii* cells were exposed to the CFCM of α-toxin mutant, a decrease in biofilm formation was observed in both *A. baumannii* strains. These results agree with the observations of Cohen et al. [12], indicating that α-toxin contributes to the proliferation and survival of *A. baumannii*.

The *A. baumannii* strains analyzed in this work are characterized by not having hemolytic activity; however, in the presence of CFCM_LS1,_ the hemolytic activity was triggered in *A. baumannii*, which adds a novel aspect of the virulence of polymicrobial infections. It should be noted that *agr*-deficient *S. aureus* did not potentiate the hemolytic activity of *A. baumannii*, as the Agr system plays a role in the enhanced hemolytic activity in *S. aureus* [37]. Like this, our previous report showed that extracellular products from non-hemolytic *S. aureus* LS1 potentiate the hemolysis of *Burkholderia cepacia* complex strains [24].

To increase our understanding of relevant aspects of the CFCM of *S. aureus* with *A. baumannii* interactions, we carried out a transcriptomic analysis comparing *A. baumannii* grown in LB versus *A. baumannii* grown in the presence of CFCM_USA300_. Our results indicated that 463 genes were differentially expressed under CFCM treatment. *A. baumannii* may be tuning its transcriptional response to survive in this condition, a trait that has been documented previously. For instance, quorum sensing allows bacteria to maintain cell–cell communication and regulate the expression of specific genes, in response to changes in cell population density [47]. There are two quorum-sensing processes described for bacteria. The type-1 auto-inducers are species-specific and are engaged for intraspecies communication, while the type-2 auto-inducers are not species-specific and are used for interspecies communication [47]. In this study, an interspecies communication system was described. The extracellular products of *S. aureus* could modify different aspects of *A. baumannii*, specifically behavior such as motility, biofilm formation, hemolytic activity, antibiotic resistance profiles and expression of virulence factors. However, we recognize that the present study possesses some limitations, such as the lack of the in vivo effect of the co-existence of *S. aureus* and *A. baumannii*. In addition, the direct role of the α-toxin or specific secondary metabolites released by *S. aureus* need to be further studied. Importantly, the present study sets the stage for further studies, which can further evidence that *A. baumannii* can perceive and respond to effectors released by *S. aureus*.

## 5. Conclusions

The present results revealed that the coexistence between bacteria can result in modifications in their general biology. *A. baumannii* can respond to soluble molecules secreted by *S. aureus*, which is in line with previous studies that show strain effects and responses. The versatility of *A. baumannii* to sense and respond to *S. aureus*‘ molecules demonstrates evidence of its exceptional ability to adapt to different conditions.

## Figures and Tables

**Figure 1 biology-11-00570-f001:**
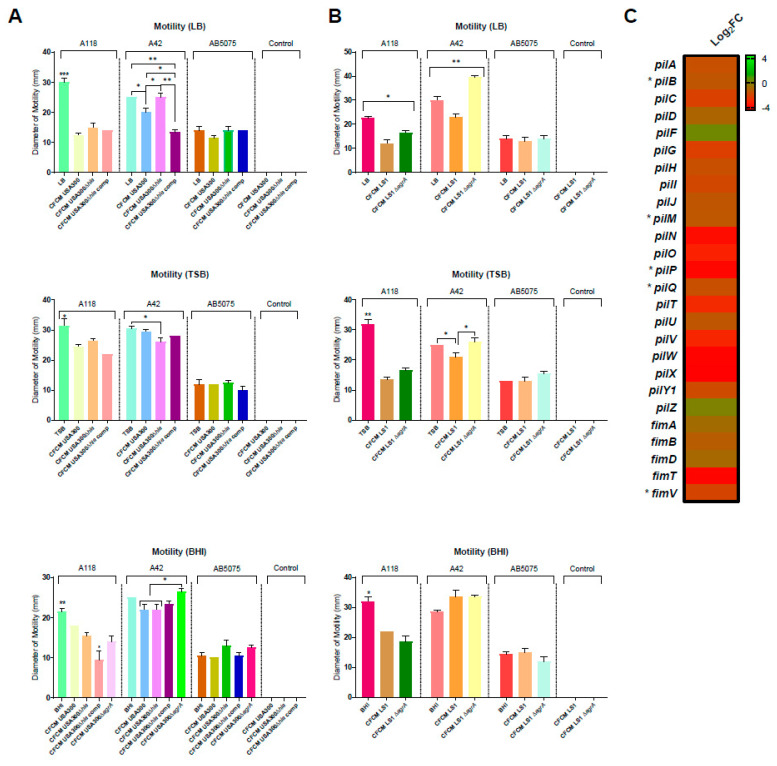
Phenotypic and transcriptomic analysis of associated motility coding genes. Diameter of motility (mm) of *A. baumannii* A118, A42 and AB5075 under CFCM_USA300_, CFCM_USA300 Δ*hla*_, CFCM_USA300 Δ*hla*_
_comp_ or CFCM_USA300 Δ*agr*_ (only in BHI medium) (**A**) or CFCM_LS1_ and CFCM_LS1 Δ*agr*_ treatments (**B**) obtained from LB, TSB and BHI broth. Experiments were performed in triplicate, with at least three technical replicates per biological replicate. The control condition corresponds to LB, TSB, or BHI supplemented with CFCM, without inoculating with *A. baumannii* strains. Statistical analysis for each strain (ANOVA followed by Tukey’s multiple comparison test) was performed using GraphPad Prism (GraphPad software, San Diego, CA, USA), and a *p*-value < 0.05 was considered significant, one asterisk: *p*-value < 0.05; two asterisks: *p*-value < 0.01 and three asterisks: *p*-value < 0.001. The asterisks observed in the graphs indicate significant differences between the condition with one or more asterisks and the rest of the conditions, except when it is indicated with brackets between these conditions. (**C**) Heatmap showing the differential expression of genes associated with fimbriae biogenesis, structural organization of T4P and motility, and type I pilus. Significantly differentially expressed genes (DEG) are indicated with an asterisk (FDR < 0.05).

**Figure 2 biology-11-00570-f002:**
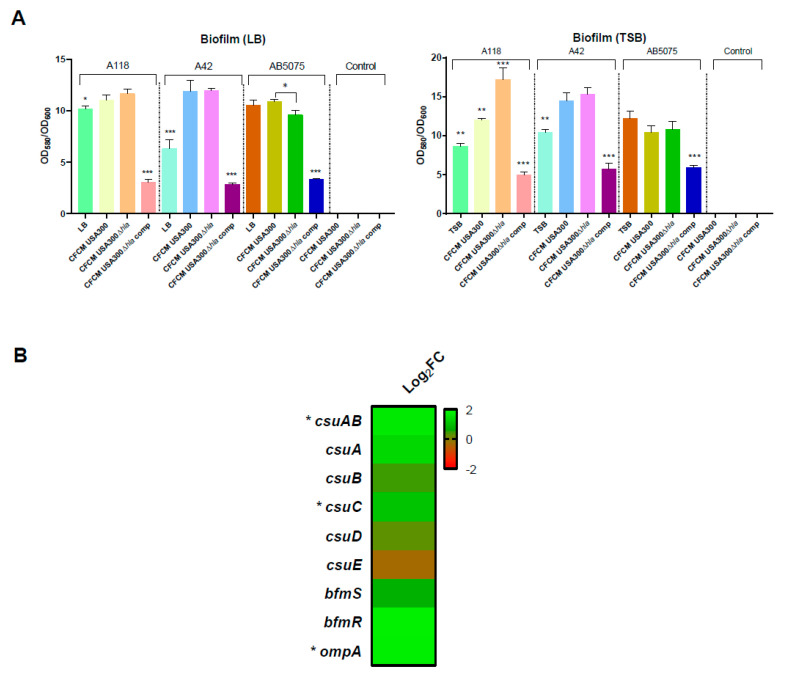
Phenotypic and transcriptomic analysis of associated biofilm formation coding genes. (**A**) Biofilm assays performed with and without CFCM_USA300_, CFCM_USA300 Δ*hla*_, or CFCM_USA300 Δ*hla*_
_comp_ obtained from LB and TSB represented by OD580/OD600. Experiments were performed in triplicate, with at least three technical replicates per biological replicate. The control condition corresponds to LB or TSB supplemented with CFCM, without inoculating with *A. baumannii* strains. Statistical analysis for each strain (ANOVA followed by Tukey’s multiple comparison test) was performed using GraphPad Prism (GraphPad software, San Diego, CA, USA), and a *p*-value < 0.05 was considered significant, one asterisk: *p*-value < 0.05; two asterisks: *p*-value < 0.01 and three asterisks: *p*-value < 0.001. The asterisks observed in the graphs indicate significant differences between the condition with one or more asterisks and the rest of the conditions, except when it is indicated with brackets between these conditions. (**B**) Heatmap showing the differential expression of genes associated with biofilm formation. DEG are indicated with an asterisk (FDR < 0.05).

**Figure 3 biology-11-00570-f003:**
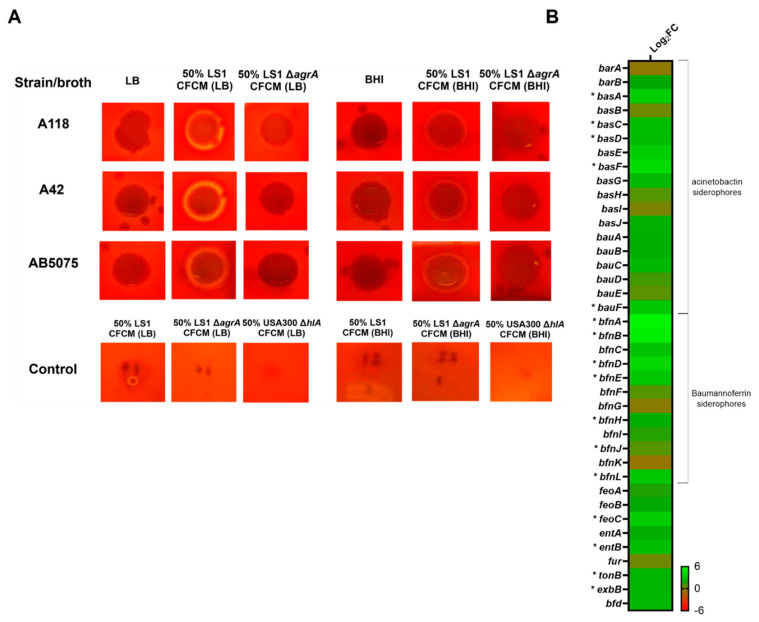
Phenotypic and genetic analysis of iron acquisition and metabolism coding genes. (**A**) Hemolytic activity was performed for *A. baumannii* A118, A42 and AB5075 strains in presence or absence of different CFCM obtained from LB or BHI broth. The control condition corresponds to LB or BHI supplemented with CFCM, without *A. baumannii* cells. (**B**) Heatmap showing the differential expression of genes associated with iron acquisition and metabolism. DEG are indicated with an asterisk (FDR < 0.05).

**Figure 4 biology-11-00570-f004:**
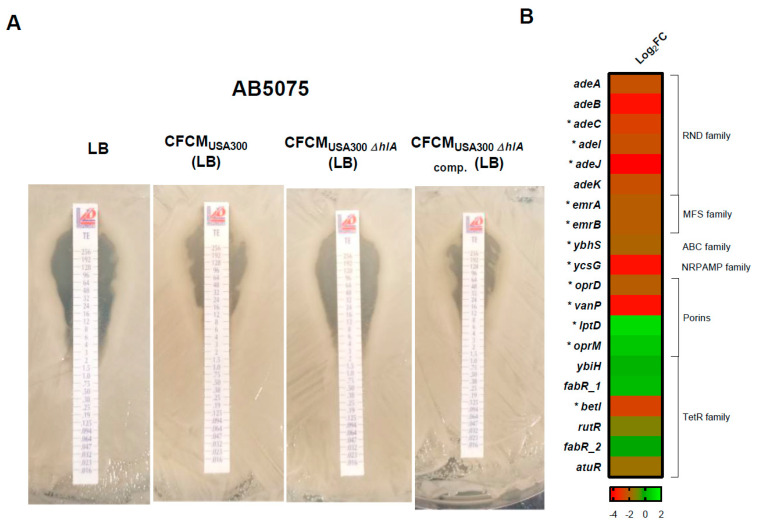
Phenotypic and transcriptomic analysis of antibiotic resistance coding genes. (**A**) Minimum inhibitory concentration (MIC) was performed by E-test (Liofilchem, Italy) following CLSI recommendations for *A. baumannii* AB5075 in presence or absence of different CFCM. (**B**) Heatmap showing the differential expression of genes associated with antibiotic resistance. DEG are indicated with an asterisk (FDR < 0.05).

## Data Availability

All data pertaining to the study described in the manuscript are described in the report.

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
