# Peer review of "Staphylococcus aureus α-Toxin Effect on Acinetobacter baumannii Behavior"

_biology, 2022, doi:10.3390/biology11040570_

Round 1
Reviewer 1 Report
I recommend following the classic structure of the article and describing Section 4 Materials and Methods before the Results section and the Discussion. To understand the results of a study, you must first become familiar with the methods.
The previously published conclusion that both strains recovered from the same site of infection, can co-exist in the same site of infection and there was no significant decrease in growth (9) should be clarified.
Author Response
March 7th, 2022
Dear Ms. Tess Wang,
We would like to thank you for your assistance with our manuscript. We are grateful to the anonymous reviewers who have helped us to improve the manuscript.
You will find enclosed the revised version of the manuscript. We have considered the reviewer’s suggestions and answered their questions. The following is a point-by-point response to the reviewer’s comments. In addition, we have added the conclusions section. We hope that now you find the manuscript suitable for publication.
Yours sincerely,
Maria Soledad Ramirez, PhD.
Assistant Professor of Biological Science
California State University Fullerton
Reviewer #1: I recommend following the classic structure of the article and describing Section 4 Materials and Methods before the Results section and the Discussion. To understand the results of a study, you must first become familiar with the methods.
Author response:
We corrected the structure of the article accordingly, Section 2 Material and Methods and Section 3 Results.
Reviewer #1: The previously published conclusion that both strains recovered from the same site of infection, can co-exist in the same site of infection and there was no significant decrease in growth (9) should be clarified.
Author response:
As suggested, we added a brief description of results obtained in Castellanos et al (1).
Castellanos N, Nakanouchi J, Yüzen DI, Fung S, Fernandez JS, Barberis C, Tuchscherr L, Ramirez MS. 2019. A Study on Acinetobacter baumannii and Staphylococcus aureus Strains Recovered from the Same Infection Site of a Diabetic Patient. Curr Microbiol 76:842-847.
Reviewer 2 Report
Well written and gives insight to adaptation capability of Acinetobacter strains. A few corrections regarding spelling: line 52 released instead of releases, line 62 assays instead of assay, lines 132, 138 increase instead of increased, line 198 sensed instead of sense
Author Response
March 7th, 2022
Dear Ms. Tess Wang,
We would like to thank you for your assistance with our manuscript. We are grateful to the anonymous reviewers who have helped us to improve the manuscript.
You will find enclosed the revised version of the manuscript. We have considered the reviewer’s suggestions and answered their questions. The following is a point-by-point response to the reviewer’s comments. In addition, we have added the conclusions section. We hope that now you find the manuscript suitable for publication.
Yours sincerely,
Maria Soledad Ramirez, PhD.
Assistant Professor of Biological Science
California State University Fullerton
Reviewer #2: Well written and gives insight to adaptation capability of Acinetobacter strains. A few corrections regarding spelling
Author response:
We appreciate the reviewer’s observation and we have updated the spelling as suggested
Reviewer #2: line 52 released instead of releases
Author response:
Changed as suggested.
Reviewer #2: line 62 assays instead of assay
Author response:
Changed as suggested.
Reviewer #2: lines 132, 138 increase instead of increased
Author response:
Changed as suggested.
Reviewer #2: line 198 sensed instead of sense
Author response:
Changed as suggested
Reviewer 3 Report
The manuscript describes the effects of the Staphylococcus aureus α-toxin with Acinetobacter baumannii and found that Acinetobacter baumannii can sense and respond to molecules secreted by Staphylococcus aureus. The manuscript is written very well. A few minor changes include:
1. Line 267-268: CLSI citation should be included with the modified procedure described by Ramirez et al.
2. The study's limitations should be included, especially regarding the lack of the in vivo effect of the co-existence of Staphylococcus aureus and Acinetobacter baumannii.
3. There are a few formatting issues throughout the manuscript that should be corrected.
Author Response
March 7th, 2022
Dear Ms. Tess Wang,
We would like to thank you for your assistance with our manuscript. We are grateful to the anonymous reviewers who have helped us to improve the manuscript.
You will find enclosed the revised version of the manuscript. We have considered the reviewer’s suggestions and answered their questions. The following is a point-by-point response to the reviewer’s comments. In addition, we have added the conclusions section. We hope that now you find the manuscript suitable for publication.
Yours sincerely,
Maria Soledad Ramirez, PhD.
Assistant Professor of Biological Science
California State University Fullerton
Reviewer #3: The manuscript describes the effects of the Staphylococcus aureus α-toxin with Acinetobacter baumannii and found that Acinetobacter baumannii can sense and respond to molecules secreted by Staphylococcus aureus. The manuscript is written very well. A few minor changes include:
Reviewer #3: Line 267-268: CLSI citation should be included with the modified procedure described by Ramirez et al.
Author response:
CLSI citation was included as suggested.
Reviewer #3: The study's limitations should be included, especially regarding the lack of the in vivo effect of the co-existence of Staphylococcus aureus and Acinetobacter baumannii.
Author response:
As suggested, we added the study’s limitation.
Reviewer #3: There are a few formatting issues throughout the manuscript that should be corrected.
Author response:
We appreciate the reviewer’s observation, and we corrected the formatting issues accordingly.
Reviewer 4 Report
The current study is in the area of interspecies bacterial interactions and in particular in relation to the host. These topics are extremely interesting in terms of science and human health. However, we regret that the study remains somewhat superficial and is not worth being accepted in its current state. The main shortcoming of the study is the lack of direct identification of distinct molecules that may affect the interaction of S. aureus with A. baumannii. Although α-toxin of S. aureus is mentioned in the introduction (line 52), there is no experimentation with recombinant α-toxin (or experiments with neutralizing antibodies against α-toxin) to show that any of the effects observed here cell-free conditioned media (CFCM) were due to α-toxin. Moreover the CFCM resulting after growth of S. aureus were not analyzed before being used on A. baumannii in terms of their content in secondary metabolites, pH, inhibitory RNAs to give some information of the effector molecule on A. baumannii. (see for example https://www.ncbi.nlm.nih.gov/pmc/articles/PMC7028967/ ). In line 57 of the introduction the authors state “Considering that evidence, we aimed to explore the effect of this α-toxin on A. baumannii behavior.”. This has not been shown directly in terms of experiments with α-toxin but indirectly with S. aureus strains lacking the respective gene (e.g. hla). However null mutants may have pleiotropic effects not only related to the missing gene, so these kind of experiments need additional verification by the α-toxin itself.
We also believe that the overall presentation could be improved, for example:
1. Lines 23 and 24: “we observed a decreased of A. baumannii A118 motility when exposed to cell-free conditioned media (CFCM) of S. aureus USA300 or LS1. However, an increase was observed when…”
We suggest instead: “By different phenotypic approaches, we observed a decrease in the motility of A. baumannii A118 after exposure to cell-free conditioned media (CFCM) of S. aureus strains USA300 or LS1. The motility A. baumannii A118 was however increased after exposure to the CFCM of S. aureus strains USA300 Δhla and S. aureus LSI ΔagrA.”.
2. Line 33: “This finding adds knowledge of the ability of A. baumannii to adapt to different conditions.”.
We suggest: “These finding show that A. baumannii may perceive and respond to changes in the culture medium imposed by S. aureus.” As mentioned earlier this is a very general conclusion.
3. The abbreviation DEG is described first in page 7 while DEG appears first in Fig 1 page 3
4. Line 90: “Twenty-six genes were analyzed by RNA-seq, as result five genes related to motility were DEG when A. baumannii A118 grown in presence of CFCM 91 from S. aureus USA300 (Fig. 1 C)”.” Please explain why you examined these genes and not others.
5. Line 96: “…treatments a major biofilm…”. Insert comma: “…treatments, a major biofilm…”.
6. Line 150: “2.3. RNA Sequencing Analysis
In total 463 significantly differentially expressed genes (DEG) were identified (FDR < 0.05), among them, 244 were down-regulated and 219 were up-regulated…”
Please state clearly what strains were exposed to what.
7. Lines 197,8 “In this work, we demonstrate that soluble molecules secreted by S. aureus can be sense by A. baumannii and trigger different pathogenic response to adapt to environmental changes.”.
This is a very general statement that does not provide any molecule as a candidate.
8. Lines 208,9. “The A. baumannii strains analyzed in this work are characterized by not having hemolytic activity; however, in presence of CFCMLS1, the hemolytic activity was trigger in A. baumannii, which add a novel aspect of the virulence of polymicrobial infection.”.
Correct observation but what are the effector molecules?
9. Lines 225-7: “In this work, a 224 interspecies communication system was described. In this system extracellular products of S. aureus could modify different aspects of A. baumannii behavior like motility, biofilm formation, hemolytic activity, antibiotic resistance profiles and expression of virulence.”.
“In this work” and “In this system” appears twice and is confusing. For example the “In this work” refers to referenced work or the current work? “In this system” refers to what?
10. Line 236 “A. baumannii strains A118 (highly susceptible), A42 (intermediately susceptible) and AB5075 (high resistance)…”
Highly susceptible to what?
Author Response
March 7th, 2022
Dear Ms. Tess Wang,
We would like to thank you for your assistance with our manuscript. We are grateful to the anonymous reviewers who have helped us to improve the manuscript.
You will find enclosed the revised version of the manuscript. We have considered the reviewer’s suggestions and answered their questions. The following is a point-by-point response to the reviewer’s comments. In addition, we have added the conclusions section. We hope that now you find the manuscript suitable for publication.
Yours sincerely,
Maria Soledad Ramirez, PhD.
Assistant Professor of Biological Science
California State University Fullerton
Reviewer #4: The current study is in the area of interspecies bacterial interactions and in particular in relation to the host. These topics are extremely interesting in terms of science and human health. However, we regret that the study remains somewhat superficial and is not worth being accepted in its current state. The main shortcoming of the study is the lack of direct identification of distinct molecules that may affect the interaction of S. aureus with A. baumannii. Although α-toxin of S. aureus is mentioned in the introduction (line 52), there is no experimentation with recombinant α-toxin (or experiments with neutralizing antibodies against α-toxin) to show that any of the effects observed here cell-free conditioned media (CFCM) were due to α-toxin. Moreover the CFCM resulting after growth of S. aureus were not analyzed before being used on A. baumannii in terms of their content in secondary metabolites, pH, inhibitory RNAs to give some information of the effector molecule on A. baumannii. (see for example https://www.ncbi.nlm.nih.gov/pmc/articles/PMC7028967/ ). In line 57 of the introduction the authors state “Considering that evidence, we aimed to explore the effect of this α-toxin on A. baumannii behavior.”. This has not been shown directly in terms of experiments with α-toxin but indirectly with S. aureus strains lacking the respective gene (e.g. hla). However null mutants may have pleiotropic effects not only related to the missing gene, so these kind of experiments need additional verification by the α-toxin itself.
Author response:
We appreciate the reviewer’s analysis.
As it is known, the type and amount of extracellular products that bacteria secrete can vary depending on the composition of the medium in which the bacteria are grown. Furthermore, the different effects of α-toxin protein purified has already been extensively studied (2-5). In this context, the aim of this study was to evaluate the potential role of α-toxin of S. aureus on A. baumannii behavior, utilizing the cell‑Free Conditioning Media (CFCM) of the wild type strain, ΔhlA strain, as well the strain that overexpresses the α-toxin. The use of CFCM, instead of purified α-toxin, allowed us to simulate more closely what occurs in in vivo conditions, in an in vitro assay since CFCM components could be necessary to improve the α-toxin effect on A. baumannii.
A follow-up project that will be submitted for publication in the near future extends the studies of CFCM. Our laboratory is evaluating the compositions of the different CFCM using mass spectrometry and high-pressure liquid chromatography techniques, with the aim of carrying out a detailed study. We would prefer to include those results in the upcoming publication.
Reviewer #4: We also believe that the overall presentation could be improved, for example:
Lines 23 and 24: “we observed a decreased of A. baumannii A118 motility when exposed to cell-free conditioned media (CFCM) of S. aureus USA300 or LS1. However, an increase was observed when…” We suggest instead: “By different phenotypic approaches, we observed a decrease in the motility of A. baumannii A118 after exposure to cell-free conditioned media (CFCM) of S. aureus strains USA300 or LS1. The motility A. baumannii A118 was however increased after exposure to the CFCM of S. aureus strains USA300 Δhla and S. aureus LSI ΔagrA.”.
Author response:
Changed as suggested.
Reviewer #4: Line 33: “This finding adds knowledge of the ability of A. baumannii to adapt to different conditions.”. We suggest: “These finding show that A. baumannii may perceive and respond to changes in the culture medium imposed by S. aureus.” As mentioned earlier this is a very general conclusion.
Author response:
Changed as suggested.
Reviewer #4: The abbreviation DEG is described first in page 7 while DEG appears first in Fig 1 page 3.
Author response:
We described the abbreviation in Fig. 1, as suggested.
Reviewer #4: Line 90: “Twenty-six genes were analyzed by RNA-seq, as result five genes related to motility were DEG when A. baumannii A118 grown in presence of CFCM 91 from S. aureus USA300 (Fig. 1 C)”.” Please explain why you examined these genes and not others.
Author response:
In Fig. 1, the motility was evaluated. In order to correlate this phenotype with the gene expression, we focused on the genes described in the literature that are related to motility modulation in A. baumannii. It has been corrected.
Reviewer #4: Line 96: “…treatments a major biofilm…”. Insert comma: “…treatments, a major biofilm…”.
Author response:
Added as suggested.
Reviewer #4: Line 150: “2.3. RNA Sequencing Analysis In total 463 significantly differentially expressed genes (DEG) were identified (FDR < 0.05), among them, 244 were down-regulated and 219 were up-regulated…” Please state clearly what strains were exposed to what.
Author response:
As suggested, we added the strain (A118) that was exposing to CFCMUSA300 to carried out the RNA sequencing analysis.
Reviewer #4: Lines 197,8 “In this work, we demonstrate that soluble molecules secreted by S. aureus can be sense by A. baumannii and trigger different pathogenic response to adapt to environmental changes.”. This is a very general statement that does not provide any molecule as a candidate.
Author response:
We corrected the sentence accordingly.
Reviewer #4: Lines 208,9. “The A. baumannii strains analyzed in this work are characterized by not having hemolytic activity; however, in presence of CFCMLS1, the hemolytic activity was trigger in A. baumannii, which add a novel aspect of the virulence of polymicrobial infection.”. Correct observation but what are the effector molecules?
Author response:
The effect of S. aureus LS1 on A. baumannii (as well as in Burkholderia cepacia Complex (Bcc), in our previous report (6) in hemolytic activity adds a novel aspect to those who showed that S. aureus secreted products suppressed the pro-inflammatory response in host cells induced by Bcc and A. baumannii. In addition, it is known that extracellular products mediate communication between bacteria, and this communication results in positive or negative interactions that modify different aspects of bacterial behavior (7, 8). In this context, and despite the efforts made by us as well as by other collaborators, we have not yet been able to elucidate which molecular effector(s) mediate this response. We are currently investigating this phenomenon.
Reviewer #4: Lines 225-7: “In this work, a 224 interspecies communication system was described. In this system extracellular products of S. aureus could modify different aspects of A. baumannii behavior like motility, biofilm formation, hemolytic activity, antibiotic resistance profiles and expression of virulence.”. “In this work” and “In this system” appears twice and is confusing. For example the “In this work” refers to referenced work or the current work? “In this system” refers to what?
Author response:
In both cases, we referenced to this manuscript. We corrected the sentence accordingly.
Reviewer #4: Line 236 “A. baumannii strains A118 (highly susceptible), A42 (intermediately susceptible) and AB5075 (high resistance)…”Highly susceptible to what?
Author response:
For this study, we set out to work with A. baumannii strains that possess different characteristics according to antibiotic resistance. A. baumannii strain A118 has been shown to be susceptible to a variety of antibiotics (9, 10), while A42 has shown to have a moderate susceptibility to antimicrobials (11) and AB5075 is known to be highly virulent and resistant to antimicrobials (12).
REFERENCES
- Castellanos N, Nakanouchi J, Yüzen DI, Fung S, Fernandez JS, Barberis C, Tuchscherr L, Ramirez MS. 2019. A Study on Acinetobacter baumannii and Staphylococcus aureus Strains Recovered from the Same Infection Site of a Diabetic Patient. Curr Microbiol 76:842-847.
- Anderson MJ, Lin YC, Gillman AN, Parks PJ, Schlievert PM, Peterson ML. 2012. Alpha-toxin promotes Staphylococcus aureus mucosal biofilm formation. Front Cell Infect Microbiol 2:64.
- Anderson MJ, Schaaf E, Breshears LM, Wallis HW, Johnson JR, Tkaczyk C, Sellman BR, Sun J, Peterson ML. 2018. Alpha-Toxin Contributes to Biofilm Formation among Staphylococcus aureus Wound Isolates. Toxins (Basel) 10.
- Todd OA, Fidel PL, Harro JM, Hilliard JJ, Tkaczyk C, Sellman BR, Noverr MC, Peters BM. 2019. Candida albicans Augments Staphylococcus aureus Virulence by Engaging the Staphylococcal. mBio 10.
- Hildebrand A, Pohl M, Bhakdi S. 1991. Staphylococcus aureus alpha-toxin. Dual mechanism of binding to target cells. J Biol Chem 266:17195-200.
- Moriano A, Serra DO, Hoard A, Montana S, Degrossi J, Bonomo RA, Papp-Wallace KM, Ramirez MS. 2021. Staphylococcus aureus Potentiates the Hemolytic Activity of Burkholderia cepacia Complex (Bcc) Bacteria. Curr Microbiol 78:1864-1870.
- Cohen TS, Hilliard JJ, Jones-Nelson O, Keller AE, O'Day T, Tkaczyk C, DiGiandomenico A, Hamilton M, Pelletier M, Wang Q, Diep BA, Le VT, Cheng L, Suzich J, Stover CK, Sellman BR. 2016. Staphylococcus aureus alpha toxin potentiates opportunistic bacterial lung infections. Sci Transl Med 8:329ra31.
- Vu B, Chen M, Crawford RJ, Ivanova EP. 2009. Bacterial extracellular polysaccharides involved in biofilm formation. Molecules 14:2535-54.
- Ramirez MS, Penwell WF, Traglia GM, Zimbler DL, Gaddy JA, Nikolaidis N, Arivett BA, Adams MD, Bonomo RA, Actis LA, Tolmasky ME. 2019. Identification of Potential Virulence Factors in the Model Strain Acinetobacter baumannii A118. Front Microbiol 10:1599.
- Traglia GM, Chua K, Centron D, Tolmasky ME, Ramirez MS. 2014. Whole-genome sequence analysis of the naturally competent Acinetobacter baumannii clinical isolate A118. Genome Biol Evol 6:2235-9.
- Vilacoba E, Almuzara M, Gulone L, Traglia GM, Figueroa SA, Sly G, Fernandez A, Centron D, Ramirez MS. 2013. Emergence and spread of plasmid-borne tet(B)::ISCR2 in minocycline-resistant Acinetobacter baumannii isolates. Antimicrob Agents Chemother 57:651-4.
- Jacobs AC, Thompson MG, Black CC, Kessler JL, Clark LP, McQueary CN, Gancz HY, Corey BW, Moon JK, Si Y, Owen MT, Hallock JD, Kwak YI, Summers A, Li CZ, Rasko DA, Penwell WF, Honnold CL, Wise MC, Waterman PE, Lesho EP, Stewart RL, Actis LA, Palys TJ, Craft DW, Zurawski DV. 2014. AB5075, a Highly Virulent Isolate of Acinetobacter baumannii, as a Model Strain for the Evaluation of Pathogenesis and Antimicrobial Treatments. MBio 5:e01076-14.
Round 2
Reviewer 4 Report
The authors have addressed the topics concerning the presentation of the manuscript but the main issue/question remains. If the effector molecule in the cell free conditioned media is α toxin, then the research is not that original. This could have been addressed by the use of neutralizing antibodies against α toxin but the experiment was not performed. It would have been promising to show that the effect remained in the absence of α toxin. A partial characterization of the suspected effector(s) (not being the toxin) would render the new manuscript worthy of publication. Regrettably, the current manuscript is not complete for acceptance.
